# YB-1 Is a Novel Target for the Inhibition of α-Adrenergic-Induced Hypertrophy

**DOI:** 10.3390/ijms25010401

**Published:** 2023-12-28

**Authors:** Jacqueline Heger, Stefan Partsch, Claudia Harjung, Zoltán V. Varga, Tamás Baranyai, Johannes Weiß, Lea Kremer, Fabian Locquet, Przemyslaw Leszek, Bence Ágg, Bettina Benczik, Péter Ferdinandy, Rainer Schulz, Gerhild Euler

**Affiliations:** 1Institute of Physiology, Justus Liebig University, 35392 Giessen, Germany; stefan.partsch@freenet.de (S.P.); c_harjung@gmx.de (C.H.); weiss.joh@icloud.com (J.W.); leakremer94@gmail.com (L.K.); fabian.locquet@gmail.com (F.L.); rainer.schulz@physiologie.med.uni-giessen.de (R.S.); gerhild.euler@physiologie.med.uni-giessen.de (G.E.); 2HCEMM-SU Cardiometabolic Immunology Research Group, 1094 Budapest, Hungary; varga.zoltan@med.semmelweis-univ.hu; 3Cardiometabolic and MTA-SE System Pharmacology Research Group, Department of Pharmacology and Pharmacotherapy, Semmelweis University, 1094 Budapest, Hungary; baranyai.tamas@med.semmelweis-univ.hu (T.B.); bence.agg@pharmahungary.com (B.Á.); bettina.benczik@pharmahungary.com (B.B.); peter.ferdinandy@pharmahungary.com (P.F.); 4Department of Heart Failure and Transplantology, Cardinal Stefan Wyszyński Institute of Cardiology, 04-628 Warszawa, Poland; p.leszek@ikard.pl; 5Pharmahungary Group, 6722 Szeged, Hungary

**Keywords:** Y-box binding protein 1, cardiomyocytes, H9C2 cells, hypertrophy

## Abstract

Cardiac hypertrophy resulting from sympathetic nervous system activation triggers the development of heart failure. The transcription factor Y-box binding protein 1 (YB-1) can interact with transcription factors involved in cardiac hypertrophy and may thereby interfere with the hypertrophy growth process. Therefore, the question arises as to whether YB-1 influences cardiomyocyte hypertrophy and might thereby influence the development of heart failure. YB-1 expression is downregulated in human heart biopsies of patients with ischemic cardiomyopathy (*n* = 8), leading to heart failure. To study the impact of reduced YB-1 in cardiac cells, we performed small interfering RNA (siRNA) experiments in H9C2 cells as well as in adult cardiomyocytes (CMs) of rats. The specificity of YB-1 siRNA was analyzed by a miRNA-like off-target prediction assay identifying potential genes. Testing three high-scoring genes by transfecting cardiac cells with YB-1 siRNA did not result in downregulation of these genes in contrast to *YB-1*, whose downregulation increased hypertrophic growth. Hypertrophic growth was mediated by PI3K under PE stimulation, as well by downregulation with YB-1 siRNA. On the other hand, overexpression of *YB-1* in CMs, caused by infection with an adenovirus encoding *YB-1* (AdYB-1), prevented hypertrophic growth under α-adrenergic stimulation with phenylephrine (PE), but not under stimulation with growth differentiation factor 15 (GDF15; *n* = 10–16). An adenovirus encoding the green fluorescent protein (AdGFP) served as the control. *YB-1* overexpression enhanced the mRNA expression of the Gq inhibitor regulator of G-protein signaling 2 (*RGS2*) under PE stimulation (*n* = 6), potentially explaining its inhibitory effect on PE-induced hypertrophic growth. This study shows that YB-1 protects cardiomyocytes against PE-induced hypertrophic growth. Like in human end-stage heart failure, YB-1 downregulation may cause the heart to lose its protection against hypertrophic stimuli and progress to heart failure. Therefore, the transcription factor YB-1 is a pivotal signaling molecule, providing perspectives for therapeutic approaches.

## 1. Introduction

Heart failure is a progressive disease associated with pathological cardiac hypertrophy that affects the heart’s contractility [1]. The prognosis for heart failure is poor, with a very high mortality rate within one year after the initial heart failure diagnosis. The body reacts with compensatory mechanisms to maintain cardiac function, such as activating the sympathetic nervous system and hypertrophic growth [2]. Despite initial heart function amelioration, hypertrophic cardiomyocyte growth predicts the development of heart failure [3]. The transcription factor Y-box binding protein 1 (YB-1) is upregulated in the early phase after myocardial infarction in rats and may contribute to post-infarct remodeling processes [4]. Interestingly, *YB-1* knockdown in mice increased their heart weight-to-body weight and heart weight-to-tibia length ratios [5]. This study attributed cardiac hypertrophy to severe fibrosis under YB-1 knockdown. However, YB-1’s influence on cardiomyocyte hypertrophy was not examined in that study. Therefore, it is pertinent to explore YB-1’s role in cardiomyocyte growth.

YB-1 is a highly conserved protein and belongs to a DNA- and RNA-binding factor family called cold-shock proteins. It is ubiquitously distributed in all tissues and contributes to various cellular functions in transcription and translation [6], splicing [7], and protein repair interactions [8]. YB-1 has three domains: a variable N-terminus, a highly conserved central nucleic-acid-binding domain (the cold-shock domain), and a hydrophilic C-terminal end [6]. The N-terminal domain promotes transactivation, and the C-terminal domain interacts with single- and double-stranded DNA and RNA [9,10] and other proteins [11]. YB-1 is mostly localized in the cytoplasm. Stimulation with interferon-gamma (IFN-γ) or transforming growth factor-beta (TGFβ) causes YB-1’s phosphorylation and translocation to the nucleus [12,13]. Here, YB-1 can regulate transcription by binding to the Y-box, most of which have an inverted CCAAT-box (ATTGG) as their core binding site [6]. This sequence exists in many eukaryotic genes. In addition to the Y-box, YB-1 can bind together with other transcription factors to different promoter regions [12]. A further mechanism is YB-1 binding to single-stranded DNA in the promoter region, anticipating the binding of other transcription factors [14,15,16]. YB-1’s numerous regulatory functions mean that it is involved in regulating cell growth, death, and fibrosis, all of which are crucial in the development of heart failure.

G-protein-coupled receptors such as α1-adrenergic receptors play an important role in the development of cardiac hypertrophy, which can be repressed by the regulator of G-protein signaling (RGS) proteins [17]. α1-Adrenergic cardiomyocyte stimulation activates early genes of the activator protein 1 (AP-1) family and reactivates fetal growth genes [18]. AP-1 was identified as a pro-hypertrophic transcription factor after α-adrenergic stimulation with phenylephrine (PE) [19]. YB-1 was found to interact with AP-1 in HeLa cells, leading to decreased expression of the AP-1 target gene matrix metallopeptidase 12 (MMP12) [20]. Therefore, YB-1 might also affect hypertrophic growth under PE stimulation. Another pro-hypertrophic factor in isolated ventricular myocardial cells is growth differentiation factor 15 (GDF15) [21], a TGFβ family member that acts not via G-protein-coupled receptors but, instead, via activation of the transcription factor SMAD family member 1 (SMAD1) in cardiomyocytes [21]. It has been described as a prognostic biomarker in patients with chronic heart failure [22]. YB-1 interferes with the TGFβ signaling pathway in several tissues by interacting with SMADs. YB-1’s inhibitory effect on the SMAD pathway suggests that it may also affect GDF15-induced hypertrophy.

In this study, we aimed to determine whether YB-1 is modified in cardiac patients who develop heart failure, and we investigated YB-1’s influence on PE- and GDF15-induced hypertrophy as a predictor of the development of heart failure. In cardiomyocytes, we downregulated YB-1 using small interfering RNAs (siRNAs) and upregulated YB-1 using an adenovirus vector. In this way, we aimed to determine whether YB-1 could act as a factor for the selective inhibition of pathological cardiac hypertrophy to protect against the development of heart failure.

## 2. Results

### 2.1. YB-1 Expression Is Downregulated in Human Heart Failure

To investigate YB-1’s role in heart failure, we analyzed left ventricular samples from patients with end-stage ICM. Actually, YB-1 protein levels were significantly decreased in ICM compared to control samples (Figure 1 and Appendix A).

### 2.2. YB-1 Silencing in H9C2 Cells Promotes Hypertrophic Growth

To examine the impact of reduced expression of *YB-1* on cardiac cells, we knocked down *YB-1* using siRNAs and investigated the effects of YB-1 deficiency on hypertrophic growth in H9C2 cells. *YB-1* mRNA levels were significantly lower in H9C2 cells transfected with *YB-1* siRNA than in non-transfected control cells after 24 h (Figure 2A). *YB-1* mRNA levels were further decreased in H9C2 cells transfected with YB-1 siRNA and differed significantly from those of H9C2 cells transfected with Ctrl siRNA or non-transfected cells after 48 h (Figure 2A). YB-1 protein levels were significantly lower in H9C2 cells transfected with YB-1 siRNA than in H9C2 cells transfected with Ctrl siRNA or non-transfected cells after 72 h (Figure 2B). Subsequently, a significant increase in the size of the H9C2 cells was observed due to the *YB-1* silencing (Figure 2C–F). The Ctrl siRNA did not affect the size of the H9C2 cells, indicating that the YB-1 siRNA’s growth-promoting effect was YB-1-specific. In control experiments with PE, YB1 siRNA showed a much greater increase in cell size than PE (Appendix A). This increase was significant for PE and Ctrl siRNA, whereas there was no change in cell size between YB-1 siRNA and YB-1 siRNA plus PE. This further indicates that the increase in cell size under YB-1 siRNA is due to hypertrophic growth.

### 2.3. YB-1 Silencing in Adult Rat Cardiomyocytes Promotes Hypertrophic Growth

To verify these results in primary cells as well, we additionally analyzed cardiac cells of adult rats. Forty-eight hours after transfection of cardiomyocytes with *YB-1* siRNA, the *YB-1* mRNA levels were significantly decreased (Figure 3A). YB-1 protein levels were also significantly decreased after 48 and 72 h compared to transfection with Ctrl siRNA or non-transfection, respectively (Figure 3B,C). While the transfection of cardiomyocytes with *YB-1* siRNA increased protein synthesis, transfection with Ctrl siRNA had no effect. Incubation of cells with the α-adrenergic agonist phenylephrine (PE) resulted in a marked increase in the protein synthesis rate compared to control cells, which was comparable to the enhanced cell growth with *YB-1* siRNA. Combining the *YB-1* siRNA with PE had no additive effect on hypertrophy. Therefore, *YB-1* silencing in primary adult cardiomyocytes induces hypertrophic growth (Figure 3D). The results of the cross sectional area (CSA) determination were similar. The α-adrenergic agonist phenylephrine (PE) and *YB-1* silencing increased the cell size of cardiomyocytes (Figure 3E), whereas Ctrl siRNA had no effect on cell size.

### 2.4. YB-1 siRNA Specifically Downregulates YB-1

To further verify the specificity of the *YB-1* siRNA, miRNA-like off-target prediction was performed to identify potential genes that could also be regulated by the *YB-1* siRNA (Figure 4). More detailed results are shown in Appendix A. Of these predicted target genes, three high-scoring genes were selected to test the specificity of *YB-1* siRNA. While the *YB-1* control was significantly downregulated by *YB-1* siRNA and not by Ctrl siRNA, neither of these three potential off-target genes showed expression changes caused by *YB-1* siRNA (Appendix A). The results of the GO enrichment analysis of the predicted off-target genes suggested the overrepresentation of various developmental processes, among others, and no clear cardiovascular dominance could be observed among the enriched biological processes (Appendix A). Therefore, it is not likely that the observed results in the YB-1 downregulation experiment were due to miRNA-like off-target effects. This indicates that *YB-1* siRNA specifically downregulated YB-1 in our cell culture system, leading to hypertrophic growth.

### 2.5. Treatment of Cardiomyocytes with the PI3K Inhibitor Ly294002 Reduced Hypertrophic Growth

Because stimulation with PE and YB-1 siRNA resulted in similar extents of hypertrophic growth and the combination of PE and YB-1 siRNA showed no additive effect, we searched for a common signaling molecule. The PI3K inhibitor Ly294002 inhibited the markedly higher protein synthesis rate in cells incubated with PE (Figure 5). Similarly, the increased protein synthesis of cardiomyocytes transfected with *YB-1* siRNA was attenuated by adding the PI3K inhibitor. In addition, the increase in the protein synthesis rate under the combination of YB-1 siRNA and PE was also blocked by the PI3K inhibitor. Thus, there were no differences between *YB-1* siRNA- and Ly294002-treated cells with or without PE stimulation, suggesting that PE and *YB-1* deficiency mediate hypertrophic growth via PI3K.

### 2.6. YB-1 Overexpression

To determine the impact of increased expression of YB-1 on cardiomyocytes, we generated adenoviruses encoding *YB-1* (AdYB-1) and *GFP* (AdGFP; the control) and infected adult rat cardiomyocytes with 1000 MOI. *YB-1* mRNA expression increased from 1.02 ± 0.27 in non-infected cardiomyocytes to 1.61 ± 0.53 after twelve hours, to 5.48 ± 1.41 after 18 h, and to 22.11 ± 10.51 after 24 h. The AdGFP-infected cells showed similar expression levels to non-infected cells (1.26 ± 0.30; Figure 6A). YB-1 protein levels showed a slight increase after 18 h (1.51 ± 0.14), rising to 4.07 ± 0.87 after 24 h, and were significantly increased after 30 h (10.66 ± 2.38) and 36 h (18.16 ± 4.17). Infection with AdGFP did not affect *YB-1* expression (1.16 ± 0.07; Figure 6B). To illustrate the expression and distribution of YB-1 in the cell, we infected cardiomyocytes with 1000 MOI AdYB-1 for 30 h, stained the cells with YB-1-specific antibodies, and examined them under a fluorescence microscope (Figure 6C–F). In isolated quiescent adult cardiomyocytes, YB-1 was regularly distributed throughout the cytosol. The negative control, not incubated with the YB-1 antibody, showed no green staining (Figure 6C). A distinct patterned cytoplasmic signal for YB-1 could be seen in the controls. These data indicate a specific distribution of endogenous YB-1 at the sarcomeres, particularly directly after cell plating and washing (Figure 6D), compared to the negative control (Figure 6C). After 30 h of culture (Figure 6E), YB-1’s distinct structure appeared to fade somewhat in the control cells. However, the distribution at the sarcomeres was maintained. Infection of cardiomyocytes with AdYB-1 confirmed a clear increase in YB-1, which was evenly distributed throughout the cell (Figure 6F).

### 2.7. Inhibition of PE-Induced but Not GDF15-Induced Hypertrophy in YB-1-Overexpressing Cardiomyocytes

The effects of *YB-1* overexpression on PE-induced hypertrophic growth were analyzed. PE stimulation caused an increase in the protein synthesis rate and cross-sectional area. When cardiomyocytes were infected with AdYB-1 seven hours before PE stimulation, the protein synthesis rate and cross-sectional area were reduced to control levels (Figure 7A,B). To examine whether *YB-1* overexpression had anti-hypertrophic effects against other stimuli, we incubated cardiomyocytes with GDF15. Stimulation with GDF15 significantly increased the protein synthesis and cell size. However, *YB-1* overexpression in cardiomyocytes did not inhibit this hypertrophic growth response to GDF15 (Figure 7C,D).

### 2.8. Induction of RGS2 under YB-1 Overexpression

We investigated whether *YB-1* overexpression’s inhibitory effect on PE-induced hypertrophy was due to signaling molecules influencing hypertrophic processes. Interestingly, simultaneous PE (but not GDF15) stimulation (Figure 8A) significantly increased the YB-1 expression in AdYB-1-transfected cardiomyocytes. RGS2, a selective Gq signaling inhibitor [17], was increased by PE. *RGS2* expression was significantly increased by *YB-1* overexpression, potentially explaining PE’s reduced hypertrophic growth effect with *YB-1* overexpression. GDF15 stimulation of cardiomyocytes, with or without *YB-1* overexpression, did not affect *RGS2* expression (Figure 8B).

## 3. Discussion

Pathophysiological hypertrophy is a predictor of heart failure, and YB-1 is a multifunctional cellular factor that can interact with pro-hypertrophic signaling at many levels. We have now identified that YB-1 is downregulated in patients with heart failure. Furthermore, we have been able to show that YB-1 downregulation provokes hypertrophic growth in cardiomyocytes, and that YB-1 overexpression significantly inhibits α_1_-adrenergic-induced hypertrophic growth. Thus, prevention of cardiac YB-1 downregulation in patients may become a therapeutic option in the treatment of pathological hypertrophy and heart failure progression.

There is growing evidence that YB-1 appears to play a critical role in the development of heart failure. Previously, in mouse hearts, an interaction of lncRNA H19 with YB-1 was demonstrated and, in this context, downregulation of YB-1 by AAV-shYB-1 resulted in cardiac remodeling [5]. In addition, lncKCND1 interacts with YB-1, which enhances the expression of YB-1. Using a mouse TAC model for the induction of cardiac hypertrophy, the authors demonstrated decreased lncKCND1 expression and a downregulation of YB-1 [23]. Silencing of YB-1 reversed the protective role of lncKCND1 in cardiac hypertrophy. Another YB-1 degradation phenomenon was found in a mouse model of streptozotocin-induced diabetic cardiomyopathy [24]. Preventing the ubiquitination and subsequent proteasomal degradation of YB-1 in those mice had cardioprotective effects.

When analyzing YB-1’s localization in cardiomyocytes isolated from adult rats, we found a distinct patterned cytoplasmic YB-1 signal in vitro. YB-1 was apparently bound to the cytoskeleton of the cardiac muscle cells. Kojic et al. [25] showed a similar binding of YB-1 to the cytoskeleton in vivo and in vitro in skeletal muscle. Myocardial hypertrophy induces changes in the cells’ protein mass, particularly sarcomere structural proteins. Therefore, the localization of YB-1 to the sarcomere could potentially influence cardiomyocyte growth.

We showed that *YB-1* knockdown with siRNA resulted in cell enlargement in both H9C2 cells and adult cardiomyocytes. This finding supports a general role of YB-1 in growth control. The similar induction of hypertrophic growth by PE and by *YB-1* suppression, alone and combination with PE, indicates that YB-1 has no additive hypertrophic growth effects and likely uses the same signaling pathway. The transcription factor AP-1 is central to the stimulation of hypertrophic growth by PE [19] and is a direct interaction partner of YB-1 [20,25]. Therefore, YB-1 knockdown may result in maximal pro-hypertrophic AP-1 activity that cannot be further increased by adding PE.

The extent to which growth is inhibited or promoted by YB-1 may depend on several factors; these include the cell type and the timing of the investigation, but also the efficiency of the downregulation of YB-1. Unlike our findings, silencing *YB-1* in tumor cells resulted in growth inhibition [26,27]. However, in cardiomyocytes, Choong et al. [5] indirectly suggested that YB-1 affects growth, since overexpression of the long non-coding RNA H19, which interacts with YB-1 and inhibits its function, induced a slight increase in cell size. In contrast to our experiments in adult cardiomyocytes, Varma et al. [28] found a significant reduction in cell size under baseline conditions, as well as after stimulation with PE, in neonatal rat cardiomyocytes (NRCMs) transfected with *YB-1* siRNA. On the other hand, Yang et al. [23] revealed a downregulation of YB-1 in AngII-stimulated neonatal mouse cardiomyocytes, indicating that the silencing of YB-1 leads to hypertrophy. Both stimulation of cardiomyocytes by AngII and alpha-adrenergic stimulation by PE lead to pathological growth via the activation of GPCRs. Varma et al. [28] and Yang et al. [23] also found differences in YB-1’s effects in vivo. Both used TAC-induced hypertrophic mouse hearts. Whereas Varma et al. [28] showed an increase in YB-1 protein levels two days after TAC, Yang et al. [23] found decreased YB-1 expression in hypertrophic heart tissues four weeks after TAC. Interestingly, YB-1 protein expression was upregulated in mice four weeks after TAC despite YB-1 knockdown in the study of Varma et al. [28], and the increase in cardiomyocyte cell size induced by TAC was not reduced in mouse hearts with YB-1 knockdown. The protective effect of YB-1 in this study was more likely caused by the reduction in fibrosis. Differences in the efficiency and timepoints of YB-1 knockdown, as well as the diverse action of YB-1 in different cell types, may be responsible for the divergent effects of YB-1 knockdown on hypertrophic growth.

Therefore, sustained hypertrophy with concomitant reduction of YB-1 appears to lead to HF, as suggested by the reduced YB-1 protein expression in the hearts of patients.

YB-1 protein levels and activation are determined by different signaling pathways and associated transcription factors that either inhibit or promote *YB-1* expression [29]. The PI3K pathway is one of the most important signaling pathways that can lead to cardiac hypertrophy, and PE activates PI3K signaling pathways via the α1-receptor, leading to hypertrophic growth of cardiomyocytes [30]. As we have shown in this study, PI3K is also involved in the hypertrophic signaling cascade under YB-1 reduction. This might be explained by growth-related mRNAs that are silenced in complex with YB-1. PI3K-Akt signaling allows some mRNAs to be released for translation [31] that are probably responsible for cardiomyocytes’ growth.

An important aspect in relation to hypertrophic growth is that YB-1 affects mRNA translation. The induction of YB-1’s interactions with the majority of mRNAs is dynamic and dependent on PI3K-Akt signaling [31]. Evdokimova et al. [31,32] concluded that the elevation of YB-1 levels leads to translational suppression of certain mRNA subgroups, whereas the activation of PI3K-Akt signaling releases them for translation. Interestingly, Varma et al. [28], who analyzed RNAs bound to YB-1, found a noticeable enrichment of transcripts encoding mRNAs involved in the PI3K signaling pathway. Downregulation of YB-1 could thus lead to a release of mRNAs that are necessary for the PI3K signaling pathway. This would confirm our findings that knockdown of YB-1 increases PI3K-dependent cardiac cell growth. Overexpression of YB-1 can lead to the inhibition of protein synthesis, specifically affecting and overcoming PI3K-mediated effects: for example, contact-induced inhibition of growth in chicken embryos [33,34].

YB-1′s suppression of hypertrophic growth is specific to α_1_-adrenergic-induced hypertrophy with PE. PE binds to G-protein-coupled α1-adrenoceptors to induce its pro-hypertrophic effect in cardiomyocytes [35]. After receptor activation, the G protein undergoes a conformational change, which can be inhibited in cardiomyocytes by the GTPase RGS2 [36]. The early upregulation of *RGS2* expression in response to the stimulation of a G_alphaq/11_-coupled receptor might serve as a negative feedback loop to facilitate the termination of receptor signaling. However, *RGS2* expression declines in response to sustained stimulation and may exaggerate pathophysiological hypertrophy and the progression of heart failure [36]. Since *RGS2* expression was upregulated by *YB-1* overexpression under PE stimulation, this may contribute to YB-1′s anti-hypertrophic effect, since RGS2 can block the PE-induced signaling cascade. Furthermore, Yang et al. [23] have recently shown that AngII-induced hypertrophic growth can also be inhibited by the overexpression of YB-1. Since the AngII receptor is also a G-protein-coupled receptor, RGS2 may also play a role here.

The finding that there are RNA-binding protein (RBP)-binding sites in RGS2 for YB-1 can support this potential mechanism of action.

Another possible mechanism of YB-1′s inhibition of PE-induced hypertrophy is a direct interaction between YB-1 and AP-1, independent of RGS2. In HeLa cells, YB-1 interacts with AP-1, leading to the decreased expression of the AP-1 target genes, such as *MMP12* and matrix metallopeptidase 13 (*MMP13*) [20,37]. Since AP-1 mediates the α_1_-adrenergic growth response of cardiomyocytes and its inhibition leads to decreased hypertrophic growth [19,38], the interaction between AP-1 and YB-1 could repress cardiomyocyte growth. Moreover, stimulation with AngII also leads to the activation of AP-1 [39], making the influence of this transcription factor in combination with YB-1 on the modulation of hypertrophy very reasonable.

In contrast to AP-1-dependent PE-induced hypertrophy, GDF15-induced hypertrophy depends on the transcription factor SMAD1 [21]. While YB-1 is known to interfere with SMAD signaling in other cell types, such as via enhanced expression of inhibitory SMAD family member 7 (SMAD7) [40] or a direct inhibitory interaction with SMAD family member 3 (SMAD3) [12], it does not appear to affect the SMAD/hypertrophy signaling pathway in cardiomyocytes. Finally, GDF15-induced hypertrophy was not reduced by *YB-1* overexpression.

In summary, YB-1 levels are decreased in the hearts of ICM patients. Thus, those patients have lost the protective effect of YB-1 against the induction of hypertrophic growth in cardiomyocytes, and this may contribute to the progression of heart failure. Therefore, future focus should be on YB-1 as a target for the inhibition of pathophysiological hypertrophy.

## 4. Materials and Methods

### 4.1. Animals

All animals were maintained under conditions that conformed to the Guide for the Care and Use of Laboratory Animals, published by the US National Institutes of Health (NIH publication no. 85-23, revised 1996). This study was approved by the Institutional Animal Care Committee of Justus Liebig University Giessen and was registered under the numbers 419_M, 469_M, and 668_M. Male Wistar rats were obtained from Janvier (Saint-Berthevin, France).

### 4.2. Cell Lines

Viral amplification used HEK293a cells obtained from Invitrogen (Thermo Fisher Scientific, Waltham, MA, USA). H9C2 cells, a European Collection of Authenticated Cell Cultures (ECAAC) rat heart myoblast cell line (no. 88092904) gifted by Prof. Dr. Rohrbach (Giessen, Germany), were used during the growth phase.

### 4.3. Human Samples

All procedures followed the ethical standards of the responsible committee on human experimentation (institutional and national), along with the Helsinki Declaration of 1975. Informed consent was obtained from all included patients according to the protocol approved by the local ethics committee (IK-NP-0021-24/1426/14). Hearts that were explanted but could not be used for transplantation for technical reasons served as controls (*n* = 8). The patients donating these hearts had no significant previous cardiac disease. They died from head injuries (subarachnoid hemorrhage, cerebral hemorrhage, or head trauma). Patients with advanced ischemic cardiomyopathy (ICM; *n* = 8) underwent cardiac transplantation, and their explanted hearts were used in this study.

Samples from the left ventricle’s free wall were obtained at the time of explantation. Tissue samples were immediately rinsed, blotted dry, and frozen in liquid nitrogen. They were kept at −80 °C until required.

### 4.4. Materials

The following reagents were purchased for this study: Medium 199 from Boehringer (Mannheim, Germany), fetal calf serum (FCS) from PAA (Linz, Austria), crude collagenase from Biochrom (Berlin, Germany), oligonucleotides from Invitrogen (Karlsruhe, Germany), Ly294002 from Sigma (Merck Bioscience, Darmstadt, Germany), PE from Sigma (Taufkirchen, Germany), and GDF15 from R&D (Wiesbaden, Germany).

### 4.5. Recombinant Adenovirus Construction

The plasmid pcDNA3.1-YB-1, containing a cDNA encoding *YB-1*, was kindly provided by Kiyoshi Higashi from the Environmental Health Science Laboratory (Sumitomo Chemical Co., Ltd.; Konohana-ku, Osaka, Japan). The cDNA was subcloned into the pENTR1A shuttle vector, resulting in pENTR1A-YB-1. The expression vector pAdYB-1 was prepared using Invitrogen’s ViraPower Adenoviral Expression System (Karlsruhe, Germany), with pAd/CMV/V5-Dest as the destination vector. Successful recombination was confirmed by polymerase chain reaction (PCR) and sequencing analyses. The plasmid pEGFP-C3, containing a cDNA for the green fluorescent protein (GFP), was kindly provided by Axel Gödecke from the Institute of Molecular Cardiology (Heinrich-Heine-University, Düsseldorf, Germany). The expression vector pAdGFP was prepared similarly. The primary adenovirus stock was created by transfecting HEK293a cells with pAdYB-1 or pAdGFP using Lipofectamine 2000 (Invitrogen, Karlsruhe, Germany). The recombinant viruses were propagated in HEK293a cells and tittered with the Adeno-X Rapid Titer Kit (Clontech, Mountain View, CA, USA). The virus was suspended in phosphate-buffered saline (pH 7.4) with 3% sucrose (5 × 10^12^ infectious units/L) and stored at −80 °C. For final transfection, the virus (AdGFP and AdYB-1) was used at a multiplicity of infection (MOI) of 1000.

### 4.6. Cell Culture Analysis

H9C2 cells were grown in Dulbecco’s modified Eagle medium supplemented with 10% FCS, 1% penicillin/streptomycin, 100 mM sodium pyruvate, and 200 mM glutamate at 37 °C, with 5% carbon dioxide. For the planned experiments, H9C2 cells were split and plated onto several small dishes. The H9C2 cells were then cultured for 24 h before transfection with *YB-1* siRNA (SI01921591; Qiagen, Hilden, Germany; target sequence: 5′-ACC AAG GAA GAC GTA TTT GTA-3′; sense: 5′-CAA GGA AGA CGU AUU UGU ATT-3′; antisense: 5′-UAC AAA UAC GUC UUC CUU GGT-3′) or AllStars Negative Control siRNA (1027281, Qiagen, Hilden, Germany; sequences not disclosed by the manufacturer) using Lipofectamine 2000 (Invitrogen, Carlsbad, CA, USA). After 24 h, cells were transfected with siRNA according to the manufacturer’s instructions in medium without FCS. After eight hours of incubation, the medium was replaced with Dulbecco’s modified Eagle medium supplemented with 1% penicillin/streptomycin, 100 mM sodium pyruvate, 200 mM glutamate, and 1% FCS. Cells were cultured at 37 °C with 5% carbon dioxide for 72 h. Then images of the cells were obtained using an Olympus IX 70 microscope (Olympus, Hamburg, Germany) at 100–200× magnification. Images were acquired with a black-and-white video camera (F-view II, Soft Imaging System; Olympus, Hamburg, Germany). The cell area was analyzed using SIS analysis software version 3.2 (Olympus, Hamburg, Germany) by framing each cell. The analysis software automatically converted the cell area. Changes in cell area were calculated as a percentage compared to controls.

Additionally, cells were harvested at the indicated times and used for real-time quantitative reverse-transcription PCR (qRT-PCR) or Western blot analyses. The size and area of the H9C2 cells were determined 72 h after transfection with *YB-1* or control siRNA cultured in media with 1% FCS.

Ventricular cardiomyocytes were isolated from 200–250 g male Wistar rats, suspended in basal culture medium, and plated on culture dishes, which were pre-incubated overnight with 4% FCS in Medium 199 as previously described [41]. The basal culture medium (CCT) was modified Medium 199 containing Earle’s salts, 2 mM L-carnitine, 5 mmol/L taurine, 100,000 IU/L penicillin, 100 mg/L streptomycin, and 10 μmol/L cytosine-β-D-arabinofuranoside (pH 7.4). Two hours after plating, the dishes were washed twice with CCT medium, resulting in cultures of ~90% quiescent rod-shaped cells on average. For *YB-1* downregulation, the cells were transfected with YB-1 siRNA (s179415; Thermo Fisher Scientific, Waltham, MA, USA) or AllStars Negative Control siRNA (1027281; Qiagen, Hilden, Germany) directly after washing, as described above. For the inhibition of PI3K, the inhibitor Ly29002 was added prior to PE stimulation. For *YB-1* overexpression, cells were infected with AdYB-1 or the control virus for 12, 24, or 36 h. For hypertrophic growth stimulation, cells were treated with PE (10 µM) or GDF15 (3 ng/mL) for 24 h.

### 4.7. MicroRNA-like Off-Target Prediction for YB-1 siRNA

Both sense and antisense YB-1 siRNA sequences were considered for an miRNA-like off-target prediction, and their combined effect was reported. For target prediction, TargetScan version 7.2 [42] was run with custom data, which included the YB-1 siRNA sequences and the three prime untranslated region (3′-UTR) and open reading frame (ORF) sequences downloaded with the use of the UCSC Table Browser tool [43] (genome assembly “Jul. 2014 (RGSC 6.0/rn6)”, “Genes and Gene Predictions” group, “Ensembl Genes” track, “3′ UTR Exons” and “CDS Exons”). The source code of the TargetScan algorithm was downloaded from the TargetScanHuman v7.2 webpage, and minor adjustments were applied to be able to run TargetScan with *Rattus norvegicus* data. Preprocessed and filtered (total context++ score <= −0.2) TargetScan predictions were used as an input interaction database for the theoretical network in miRNAtarget™ software version 2.0 (mirnatarget.com; Pharmahungary, Szeged, Hungary) [44,45,46,47,48,49,50,51] to combine the effects of the two sequences and to construct an siRNA–target interaction network. Off-target scores were calculated by summing the total context++ scores of the siRNA–target interactions in which the target was predicted to be involved. Network visualization was performed using the EntOptLayout plugin [52] (version 2.1) for the Cytoscape software platform [53] (version 3.7.2), and important targets were further highlighted with graphical settings. Functional analysis of the predicted targets was performed by running Gene Ontology (GO) enrichment analysis [54,55]. Statistical overrepresentation testing of the PANTHER Classification System [56] version released on 11 July 2019, accessed through the Gene Ontology Consortium homepage, was applied to biological processes with default statistical test settings (Gene Ontology database version released on 9 December 2019).

### 4.8. Real-Time qRT-PCR

Total RNA was extracted from cardiomyocytes with TRIzol (Invitrogen, Karlsruhe, Germany), as described by the manufacturer. Reverse-transcription (RT) reactions were performed for 1 h at 37 °C with a final volume of 10 µL, using 1 µg of RNA, 100 ng of oligo(dT)_15_, 1 mmol/L dNTPs, 8 units of RNasin, and 60 units of Moloney murine leukemia virus reverse transcriptase. Aliquots (1.5 µL) of the synthesized cDNA were used for PCR in a final volume of 10 µL, containing primer pairs at 1.5 µmol/L, 0.4 mmol/L dNTPs, 1.5 mmol/L magnesium chloride, and 1 U of Taq polymerase. The following primers were used: *YB-1* forward, 5′-TTC GCA GTG TAG GAG ATG GA-3′; *YB-1* reverse, 5′-CTA CGA CGT GGA TAG CGT CT-3′; hypoxanthine phosphoribosyltransferase (*HPRT1*) forward, 5′-CCA GCG TCG TGA TTA GTG AT-3′; *HPRT1* reverse, 5′- CAA GTC TTT CAG TCC TGT CC-3′; *RGS2* forward, 5-AGC AAA TAT GGG CTT GCT GCA T-3′; *RGS2* reverse, 5′-GCC TCT TGG ATA TTT TGG GCA ATC-3′. The annealing temperature and cycle number resulting in a linear amplification range were explored for each assayed gene. Real-time qRT-PCR was performed in an automated thermal cycler and detected with the Bio-Rad Detection System (Hercules, CA, USA) using SYBR Green fluorescence for quantification. The results were calculated according to the 2^−ΔΔCt^ method, as previously described [57]. After the amplification reaction, the products were controlled and separated on 2% agarose gels, stained with SYBR Safe, and imaged under UV illumination.

### 4.9. Western Blots

Human biopsy homogenization used 30–45 mg of left ventricular human heart tissue in 1× RIPA buffer (Cell Signaling; NEB, Frankfurt a.M., Germany) supplemented with a protease inhibitor (Roche, Mannheim, Germany) and sodium fluoride (Sigma-Aldrich, Taufkirchen, Germany), along with a Qiagen Tissue Lyser. Homogenized tissues were centrifuged at 10,000 rcf and 4 °C for 10 min, and their supernatants were aliquoted and stored at −80 °C. Then, 14 µg of protein was loaded on 4–15% Mini-PROTEAN TGX gels and transferred onto a polyvinylidene fluoride membrane. After blocking with 5% milk in Tris-buffered saline with 0.05% Tween 20 for two hours at room temperature, blots were incubated with YB-1 (1:1000; ab12148; Abcam, Cambridge, MA, USA) and glyceraldehyde 3-phosphate dehydrogenase (GAPDH) (1:5000; CS5174; Cell Signaling, Frankfurt a.M., Germany) antibodies overnight at 4 °C. Protein bands were detected by incubation with horseradish-peroxidase-labeled anti-rabbit antibodies (1:2500 and 1:10,000; Cell Signaling, Frankfurt a.M., Germany) for two hours and at room temperature, followed by visualization with the ECL detection system (Pierce^;^ Thermo Fisher Scientific, Waltham, MA, USA).

Cardiomyocytes or H9C2 cells were homogenized by lysing them in RIPA buffer (50 mmol/L Tris hydrochloride (pH 7.5), 150 mmol/L sodium chloride, 1% Nonidet P-40, 0.5% deoxycholate, and 0.1% sodium dodecyl sulfate (SDS)) containing a protease inhibitor (1 mol/L phenylmethylsulfonyl fluoride, 1 mol/L ethylenediaminetetraacetic acid (EDTA), and 1 mg/L pepstatin). Protein content was determined using the Lowry assay [58]. Samples were denatured in Laemmli buffer at 90 °C for 5 min, loaded onto 10% SDS or NuPAGE 10% Bis-Tris gels, and transferred onto a nitrocellulose membrane. Protein levels were detected using primary antibodies against YB-1 (1:1000; ab122148; Abcam, Cambridge, USA), vinculin (VCL; 1:1000; 5% bovine serum albumin; Sigma-Aldrich, Taufkirchen, Germany), and actin (1:2000; Sigma, Taufkirchen, Germany) overnight at 4 °C. After washing, the blots were incubated with secondary horseradish-peroxidase-labeled anti-rabbit (1:2500 and 1:10,000; Cell Signaling, Frankfurt a.M., Germany) or anti-mouse (1:2500 and 1:10,000) antibodies for two hours at room temperature.

Quantification of Western blots was performed with Quantity One software version 4.6.9 (Bio-Rad, Hercules, CA, USA) using volume tools for volume analysis. The background was subtracted and the volume of the bands of YB-1 was set to the volume of the bands of the reference protein. For the presentation, all individual values were related to the mean value of the controls. The presentation is shown as a percentage.

### 4.10. Cardiomyocytes Monitoring and Cell Size Determination

Cells were viewed under an Olympus IX 70 fluorescence microscope fitted with a GFP filter cube using a 40× objective (UApo/340, NA 1.35; Olympus, Hamburg, Germany). Images were acquired with a black-and-white video camera (F-view II, Soft Imaging System; Olympus, Hamburg, Germany). Rod-shaped cardiomyocytes were randomly chosen, and their width/diameter was determined at their widest point using SIS analysis software version 3.2 (Olympus, Hamburg, Germany). The cardiomyocytes’ cross-sectional area was calculated according to the following formula: radius^2^ × π [59]. The H9C2 cells’ size was determined by measuring their area.

### 4.11. Protein Synthesis Rate Determination

The protein synthesis rate was determined by measuring the incorporation of phenylalanine in cultures exposed to L-^14^C-phenylalanine (0.1 µCi/mL) for 24 h. The incorporation of radioactivity into acid-insoluble cell mass was determined as previously described [60].

### 4.12. Statistics

All n-numbers are biological replicates. The real-time qRT-PCR was performed in duplicate, and the mean value was used for the calculation. The results are expressed as the mean ± standard error of the mean (SEM) or as means ± confidence intervals (CIs) for real-time qRT-PCR. All variables were evaluated for normal distribution using the Kolmogorov–Smirnov and Shapiro–Wilk tests. Levene’s test was used to control for variance of homogeneity. If two variables were compared directly, an unpaired two-tailed *t*-test or the Mann–Whitney test (depending on the normality of the samples) was used. For more than two variables, one-way analysis of variance (ANOVA) was used, followed by the Student–Newman–Keuls test for post hoc analyses if suitable [61]. All results with *p* < 0.05 were considered statistically significant. All data analyses were performed using SPSS software (version 27; SAS Institute Inc., Cary, NC, USA).

## Figures and Tables

**Figure 1 ijms-25-00401-f001:**
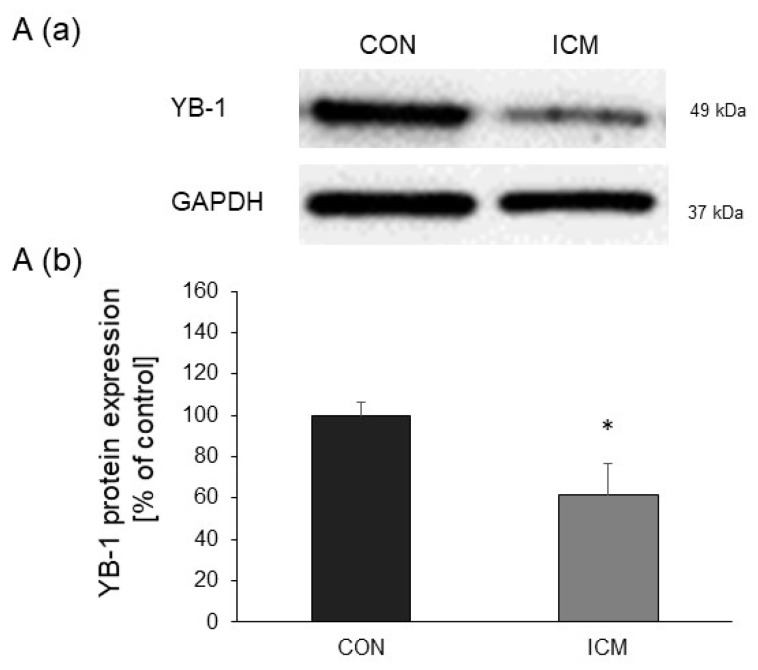
Low YB-1 protein levels in cardiac biopsies from ICM patients: YB-1 protein levels were examined with Western blots, using GAPDH as the internal control. (**A**(**a**)) Representative Western blot and (**A**(**b**)) the quantitative analysis of YB-1 protein levels (*n* = 8 control biopsies, *n* = 8 ICM biopsies). Data are presented as the means ± SEMs of unpaired two-tailed *t*-tests; * *p* < 0.05.

**Figure 2 ijms-25-00401-f002:**
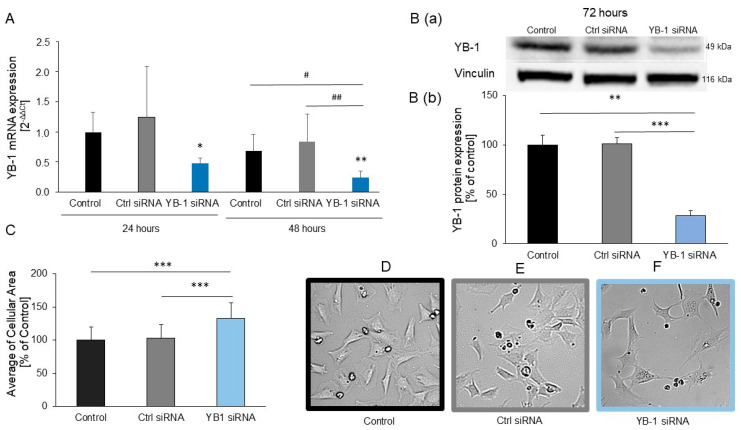
*YB-1* downregulation in H9C2 cells resulted in hypertrophic growth: H9C2 cells were transfected with *YB-1* or Ctrl siRNA or were non-transfected. (**A**) Their mRNA levels were quantified after 24 h (*n* = 4) or 48 h (*n* = 8) with real-time qRT-PCR, using *HPRT1* as the housekeeping gene. Data are presented as the means ± CIs. Key: * *p* < 0.05 versus control after 24 h; # *p* < 0.05, ## *p* < 0.01 comparison as indicated. YB-1’s protein levels were examined using Western blots (*n* = 4) after 72 h, using vinculin as the internal control. (**B**(**a**)) Representative Western blots. (**B**(**b**)) Quantitative analysis of YB-1 protein levels (*n* = 4). (**C**) Areas were measured for each cell 72 h after transfection with siRNA (*n* = 3 cell preparations, 18 dishes, with a total of 276–326 cells). (**D**–**F**) Representative images of H9C2 cells 72 h after transfection with siRNA. Data are presented as the means ± SEMs of unpaired two-tailed *t*-tests. Key: * *p* < 0.05, ** *p* < 0.01, *** *p* < 0.001.

**Figure 3 ijms-25-00401-f003:**
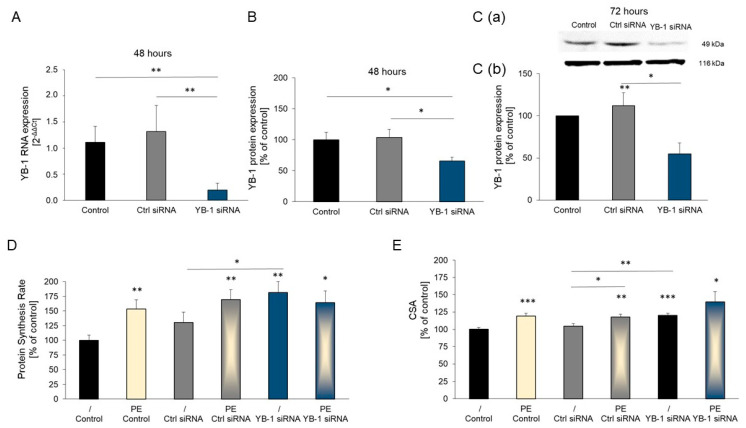
*YB-1* downregulation in adult rat cardiomyocytes induces hypertrophic growth: Cardiomyocytes were transfected with *YB-1* siRNA or Ctrl siRNA or were non-transfected. (**A**) RNA expression was quantified after 48 h (*n* = 5) with real-time qRT-PCR, using *HPRT1* as the housekeeping gene. Data are presented as the means ± CIs. (**B**) YB-1 protein levels were examined with Western blots (*n* = 6) after 48 h, using vinculin as the internal control. (**C**) YB-1 protein levels after 72 h: (**C**(**a**)) representative Western blot, and (**C**(**b**)) quantitative analysis of YB-1 protein levels (*n* = 5). (**D**) The protein synthesis rate was measured 48 h after transfection with siRNA, preceded by 24 h of PE (10 µM) stimulation (*n* = 34–37 cardiomyocyte preparations). (**E**) The cross-sectional area (CSA) of adult cardiomyocytes was measured 48 h after transfection with siRNA, preceded by 24 h of PE (10 µM) stimulation (*n* = 12 cardiomyocyte preparations). Data are presented as the means ± SEMs. Key: * *p* < 0.05, ** *p* < 0.01, *** *p* < 0.001, versus control or comparison as indicated.

**Figure 4 ijms-25-00401-f004:**
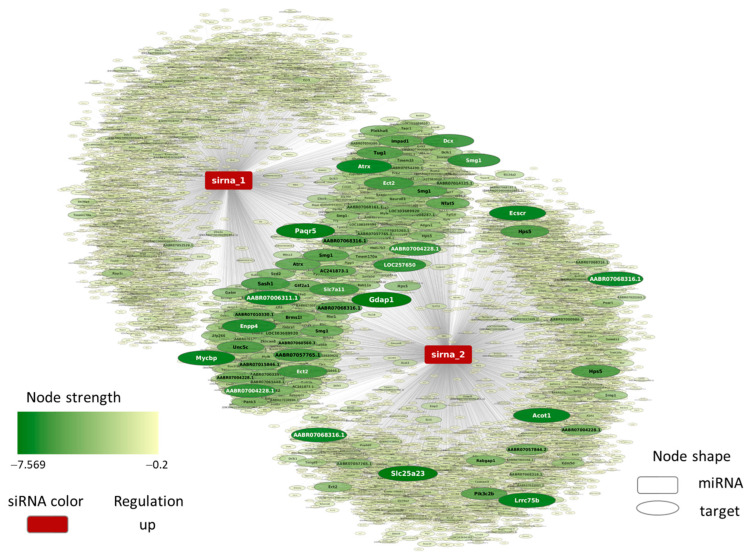
*Predicted* siRNA off-target interaction network: SiRNAs and mRNAs are presented as rectangular and oval-shaped nodes, respectively. Node size and color intensity represent the magnitude of the off-target score values. The higher the absolute value of the off-target score, the bigger and darker the node is. sirna_1: antisense strand of the siRNA, sirna_2: sense strand of the siRNA.

**Figure 5 ijms-25-00401-f005:**
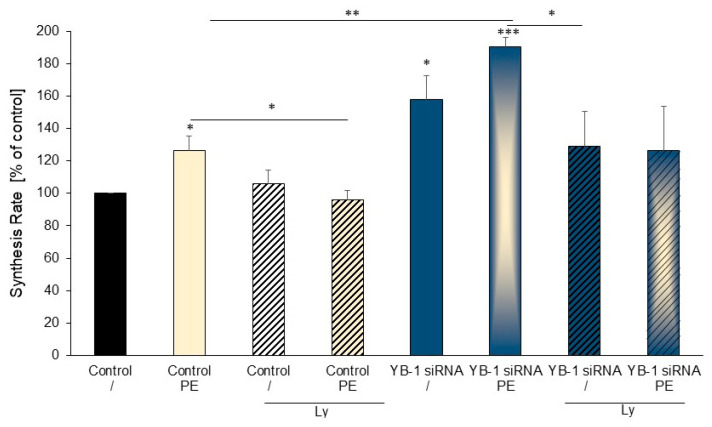
Inhibition of PI3K reduced hypertrophic growth: Cardiomyocytes were transfected with *YB-1* siRNA or were not transfected. The protein synthesis rate was measured 48 h after transfection with siRNA, preceded by 24 h of PE (10 µM) stimulation (*n* = 5 cardiomyocyte preparations). Data are presented as the means ± SEMs. Key: * *p* < 0.05, ** *p* < 0.01, *** *p* < 0.001, compared to control cells or as indicated by the bar.

**Figure 6 ijms-25-00401-f006:**
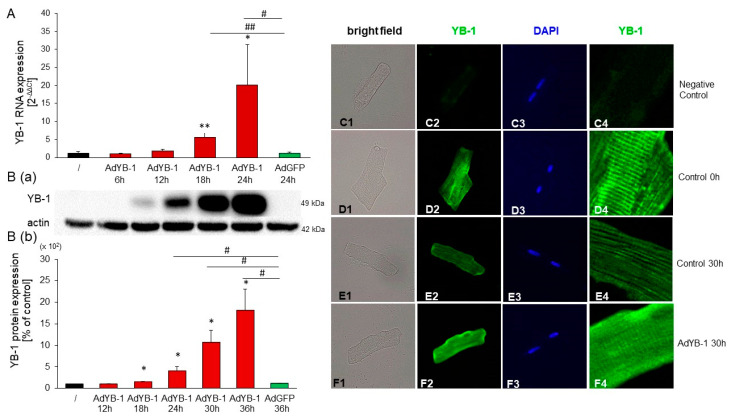
Infection of adult rat cardiomyocytes with AdYB-1 resulted in a considerable increase in *YB-1* mRNA and protein levels: (**A**) Cardiomyocytes were infected with AdYB-1 (1000 MOI) for 6, 12, 18, and 24 h and with AdGFP (1000 MOI) for 24 h or were left uninfected for 24 h. Total RNA was prepared, and *YB-1* mRNA levels were analyzed using real-time qRT-PCR, with *HPRT1* as the housekeeping gene. Data are presented as the means ± CIs across five independent culture preparations. Key: * *p* < 0.05, ** *p* < 0.01 compared to non-stimulated cardiomyocytes; # *p* < 0.05, ## *p* < 0.01, compared to AdGFP-infected cardiomyocytes. (**B**) Cardiomyocytes were infected with AdYB-1 (1000 MOI) for 12, 18, 24, 30, and 36 h and with AdGFP (1000 MOI) for 36 h or were left uninfected for 36 h. Protein extracts were prepared and run on 10% gels. YB-1 protein was detected with a YB-1-specific antibody, using actin as the loading control. The top panel (**B**(**a**)) shows a representative Western blot, while the bottom panel (**B**(**b**)) shows the quantitative analysis. Data are presented as means ± SEMs across six independent culture preparations. Key: * *p* < 0.05 compared to non-stimulated cardiomyocytes; # *p* < 0.05 compared to AdGFP-infected cardiomyocytes. (**C1**–**F4**) Cardiomyocytes were infected with AdYB-1 (1000 MOI) or left uninfected for 30 h. Immunocytochemistry analysis of the YB-1 protein was performed with a YB-1-specific antibody and an FITC-labeled secondary antibody: 1. Light exposure of single cardiomyocytes; 2. Immunofluorescence with the YB-1 antibody at 549–563 nm; 3. Nuclear staining with DAPI at 358–463 nm; 4. Immunofluorescence with the YB-1 antibody at 549–563 nm. (**C1**–**C4**) Negative control comprising non-infected cardiomyocytes without primary antibody. (**D1**–**D4**) Non-infected cardiomyocytes after plating and washing (0 h). (**E1**–**E4**) Non-infected cardiomyocytes after 30 h. (**F1**–**F4**) Cardiomyocytes infected with Ad-YB-1 (1000 MOI) after 30 h.

**Figure 7 ijms-25-00401-f007:**
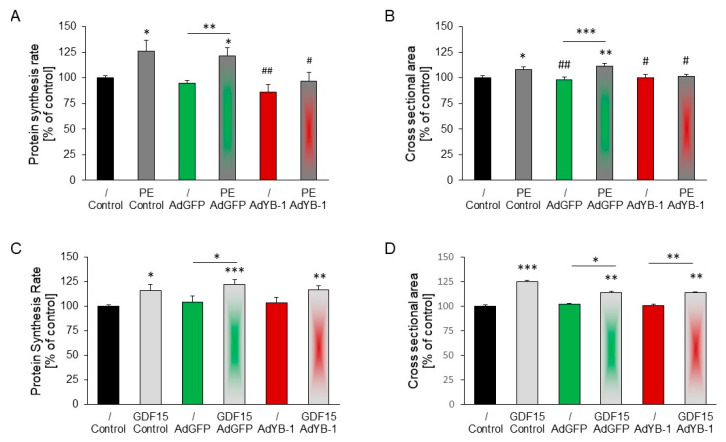
Infection of cardiomyocytes with AdYB-1 blocked PE-induced but not GDF15-induced hypertrophy: (**A**,**B**) Cardiomyocytes were infected with AdGFP (1000 MOI) or AdYB-1 (1000 MOI) for 7 h and then incubated with PE (10 µM) for 24 h. (**A**) The protein synthesis rate was determined to detect hypertrophic growth. Data are presented as means ± SEMs across 10–13 independent preparations. (**B**) The cross-sectional area was also determined. Data are presented as means ± SEMs across 371–475 cardiomyocytes from 14 independent preparations. Key: * *p* < 0.05, ** *p* < 0.01, *** *p* < 0.001, compared to unstimulated controls or between two bars marked with a dash; # *p* < 0.05, ## *p* < 0.01, compared to PE-stimulated cells. (**C**,**D**) Cardiomyocytes were infected with AdGFP (1000 MOI) or AdYB-1 (1000 MOI) for 7 h and then incubated with GDF15 (3 ng/mL) for 24 h. (**C**) The protein synthesis rate was determined to detect hypertrophic growth. Data are presented as means ± SEMs across 14–16 independent preparations. (**D**) The cross-sectional area was also determined. Data are presented as means ± SEMs across 618–709 cardiomyocytes from 11 independent preparations. Key: * *p* < 0.05, ** *p* < 0.01, *** *p* < 0.001, compared to unstimulated controls or between two bars marked with a dash.

**Figure 8 ijms-25-00401-f008:**
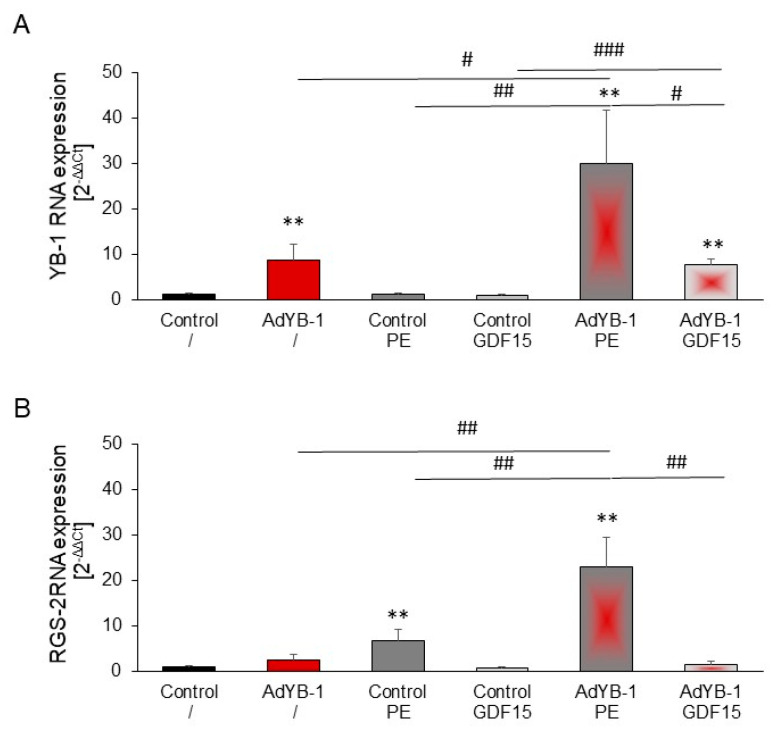
Increased RGS-2 mRNA levels in YB-1- but not GDF15-stimulated cardiomyocytes: Cardiomyocytes were infected with AdYB-1 (1000 MOI) for 7 h and then incubated with PE (10 µM) or with GDF15 (3 ng/mL) for 24 h. Total RNA was prepared, and (**A**) *YB-1* and (**B**) *RGS2* mRNA levels were determined by real-time qRT-PCR, using *HPRT1* as the housekeeping gene. Data are presented as the means ± CIs across 4–6 independent cardiomyocyte preparations, using ANOVA and unpaired two-tailed *t*-tests. Key: ** *p* < 0.01 compared to unstimulated controls; # *p* < 0.05, ## *p* < 0.01, ### *p* < 0.01, comparisons as indicated.

## Data Availability

The data presented in this study are available upon request from the corresponding author.

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
