# Peer review of "YB-1 Is a Novel Target for the Inhibition of α-Adrenergic-Induced Hypertrophy"

_ijms, 2023, doi:10.3390/ijms25010401_

Round 1
Reviewer 1 Report
Comments and Suggestions for Authors
The manuscript by Heger et al. (ijms-2674099) focused on the transcription factor Y-box binding protein 1 (YB-1) that can interact with transcription factors involved in cardiac hypertrophy and may thereby interfere with the hypertrophy growth process. YB-1 protects cardiomyocytes against PE-induced hypertrophic growth. Like in human end-stage heart failure, YB-1 downregulation may cause loss of heart protection against hypertrophic stimuli and progress to heart failure. Therefore, the transcription factor YB-1 is a pivotal signaling molecule, providing perspectives for therapeutic approaches. The manuscript is interesting, but some points need to be revised. Please find below some suggestions for the authors:
Major comments:
The authors transfected with YB-1 siRNA at different time points. The difference between the different points is not so evident. The author should add the statistical analysis also between the different time points.
The authors show an increase in the size of H9C2 due to the YB-1 silencing. An assessment of markers of cardiac hypertrophy should corroborate this finding.
Since the authors show the most evident effect on silencing YB-1 at 72 h, why did the authors check the synthesis rate with an inhibitor of PI3K at 24h?
The authors infected adult rat cardiomyocytes with 100 MOI. Why did the authors choose that concentration? And what about using different concentrations at the same time point?
The authors should add a histogram for the analysis of fluorescence.
The discussion is verbose and can be substantially reduced to a list of previous studies.
Minor comments:
Language needs to be revised.
Figure 1 The cropped image needs to be cut better. Moreover, writing A (i) and (ii) is better since it is the representative image and the quantification. Add the original blot that corresponds to the figure.
The author should add the molecular size to all representative blots.
Figure 2 Add the representative blot to 2A and 2B.
Figure 2 for C and D do the same as Figure 1, so C (i) and C(ii), and for the histograms, use the same scale as 2A and 2B
Figure 2H, please add a better-quality image.
Figure 3 C add (i) and (ii) and use the same scale as A and B
In the legend of Figure 3, there is written about a representative blot after 48 hours, but there is no image in the figure
Figure 6 B: why did the authors choose actin and not vinculin as in the other blots?
Add a better quality of images to the original blot pdf
The authors should add the dots that represent the individual values to the histograms.
The authors should add a section about abbreviations (as ICM)
Comments on the Quality of English LanguageSee the main comments.
Author Response
Answer to Reviewer 1
We thank the editor and the reviewers for their time and their constructive critiques and suggestions to improve our manuscript. We have addressed each of the comments and have revised the manuscript accordingly. We hope that the editor and reviewers will now find the revised manuscript acceptable for publication.
Major comments:
The authors transfected with YB-1 siRNA at different time points. The difference between the different points is not so evident. The author should add the statistical analysis also between the different time points.
Thank you very much for this suggestion. We have implemented your request. In contrast to the original Figures 2 A + 2B, in which we analyzed and presented the YB-1 expression after 24 and 48 hours of siRNA transfection separately, in the new Figure 2 A we now show the combined data from 24 and 48 hours of siRNA transfection. This enabled us to supplement the statistical analysis between the different points. As preferred by the other reviewer, we now present real-time qRT-PCR results as relative expression (2-ΔΔCt) and not as a percentage calculation of the relative expression.
Figure 2. YB-1 downregulation in H9C2 cells resulted in hypertrophic growth. H9C2 cells were transfected with YB-1 or Ctrl siRNA or were non-transfected. (A) Their mRNA levels were quantified after 24 hours (n = 4) or 48 hours (n = 8) with real-time qRT-PCR using HPRT1 as the housekeeping gene. Data are presented as the means ± CI. Key: *, p < 0.05 versus control after 24 hours; #, p < 0.05 comparison as indicated.
The authors show an increase in the size of H9C2 due to the YB-1 silencing. An assessment of markers of cardiac hypertrophy should corroborate this finding.
As requested by another reviewer, we have modified Figure 2 to show cell size as a percentage of control. As can be seen in this figure, the increase in cell size is just 30%. We were not able to detect any hypertrophy markers on RNA level. However, this also applies to samples with PE stimulation (which we have not shown in the manuscript). Besides, the timing of the analysis determines when which protein is expressed. BNP, which is also expressed in ventricular cardiomyocytes in contrast to ANP, is an early response gene (Nakagawa et al., JCI 1995, 1280-1287). At the times when we isolated RNA, we could no longer detect an increase in BNP.
Since the authors show the most evident effect on silencing YB-1 at 72 h, why did the authors check the synthesis rate with an inhibitor of PI3K at 24h?
The downregulation of the YB-1 protein occurs later than the downregulation of the YB-1 RNA. A decrease in YB-1 protein is visible after 48 hours at the earliest and is halved after approx. 72 hours. Adult cardiomyocytes cannot be cultivated for much longer than 72 hours without adding serum to the cultured medium. However, this changes their structure and FCS contains diverse growth stimulating substances, which would additionally modulate the hypertrophic growth responses. Therefore, the cells were first transfected with siRNA. After 48 hours, PE was added to the cells for 24 hours. For inhibition of PI3K the inhibitor Ly294002 was added after 48 hours siRNA transfection and prior to PE stimulation. This allows us to achieve the same endpoint for the determination of the protein biosynthesis rate.
The authors infected adult rat cardiomyocytes with 1000 MOI. Why did the authors choose that concentration? And what about using different concentrations at the same time point?
When we produced the virus, we tested different concentrations as well as different time intervals. We infected adult rat cardiomyocytes with 100, 500 and 1000 MOI for 24 hours as you can see in the following figure.
100 MOI was not sufficient for the infection of adult cardiomyocytes to cause a significant change compared to the control virus AdGFP (see figure above). In the end, we selected the 1000 MOI and carried out all experiments with this concentration.
The authors should add a histogram for the analysis of fluorescence.
We show the quantification of YB-1 overexpression on RNA and protein expression. We have not analyzed the intensity of the cells and therefore do not show any quantification of the fluorescence. The images are intended to show YB-1 protein localization. Therefore, we only have some representative pictures. To illustrate this even better, we have implemented a section of the cardiomyocytes as an enlargement in Figure 6 (C4-F4). This shows the localization of YB-1 at the sarcomeres, particularly in the controls. Due to the overexpression of YB-1, the localization of YB-1 can also be seen diffusely in the cytosol.
The discussion is verbose and can be substantially reduced to a list of previous studies.
The discussion has been shortened
Minor comments:
A native speaker has cross-read the manuscript for language errors.
Figure 1 The cropped image needs to be cut better. Moreover, writing A (i) and (ii) is better since it is the representative image and the quantification. Add the original blot that corresponds to the figure.
We have modified Figure 1 according to the reviewer's suggestions. The labeling was changed from A and B to A (i) and A (ii). We have chosen a different location in the original blot to display the representative blot, which better reflects the different expression levels between CON and ICM. We added the molecular size for YB-1 and GAPDH in figure A (i). The original blots of figure 1are now added in the supplement (figure S1)
The author should add the molecular size to all representative blots.
We added the molecular size to all representative blots.
Figure 2 Add the representative blot to 2A and 2B.
Figures 2 A and 2 B are quantifications of mRNA expression, so we cannot add representative blots.
Figure 2 for C and D do the same as Figure 1, so C (i) and C(ii), and for the histograms, use the same scale as 2A and 2B
We have modified Figure 2 according to the reviewer's suggestions. The labeling was changed from C and D to C (i) and C (ii).
Figure 2H, please add a better-quality image.
We have chosen a better quality image. Figure 2H is now Figure 2G.
Figure 3 C add (i) and (ii) and use the same scale as A and B
We have modified Figure 3 according to the reviewer's suggestions. The labeling was changed from C and D to C (i) and C (ii).
In the legend of Figure 3, there is written about a representative blot after 48 hours, but there is no image in the figure.
The indication for the representative Western blot refers to protein expression after 72 hours. We did not show the representative blots in the manuscript for 48 hours. For better understanding, we now rephrased the figure legend to: “(B) YB-1 protein levels were examined with western blots (n = 6) after 48 hours using vinculin as the internal control. (C) YB-1 protein levels after 72 hours: (C (i)) representative western blot, and (C (ii)) quantitative analysis of YB-1 protein levels (n = 5)”.
Western blots after 48 hours include not only samples for control, Ctrl siRNA and YB-1siRNA. The application of the samples in this blot was done in such a way that it is not possible to cut out these three samples together. We show the original blots in the supplement.
Figure 6 B: why did the authors choose actin and not vinculin as in the other blots?
This is to be seen historically. The overexpression experiments were carried out at a completely different time. At this time, actin was used by the graduate student as a reference protein in the Western blots.
Add a better quality of images to the original blot pdf
We have reassembled the PDF file for the Original blots so that it is more concise.
The authors should add the dots that represent the individual values to the histograms. The authors should add a section about abbreviations (as ICM).
The following abbreviations are now on page 18.
ICM: ischemic cardiomyopathy
PE: phenylephrine
YB-1: Y box binding protein 1
GDF15: growth differentiation factor 15
IFN-g: interferon-gamma
TGFβ: transforming growth factor-beta
RGS: the regulator of G protein signaling
The PDF file also contains figures.

Reviewer 2 Report
Comments and Suggestions for Authors
In this manuscript, the Authors claim that the downregulation of the pleiotropic YB-1 protein mediates hypertrophic growth in cardiomyocytes which can in turn lead to heart failure. While intriguing, I noticed profound technical and biological flaws which force me to reject this manuscript.
I am concerned by the modality in which the experiments were made and how they were discussed/presented in the text. Methods are incomplete and there is great confusion in the application of the statistical analysis. Plots and graphs of the very same technique (RT-qPCR for instance) are presented with a different notation for each experiment, while no indications are described in the methods.
The resolution and quality of the Original western blot file are really low and the document itself is really difficult to understand as the Authors have not indicated correspondence between figures in the text and the “Original western blot file”. Some original images are missing (Figures 2C, 3C for instance).
I am not convinced by the data showing the hypertrophic growth upon YB-1 KD, which constitute the starting point and the backbone of the story presented by the Authors.
In the first part of the manuscript the Authors put great efforts in showing the efficiency (and the absence of miRNA-mediated off-target effects) of YB-1 KD in rat cells, while data regarding the YB-1-mediated hypertrophic growth are insufficient.
No immunohistochemical, biochemical or transcriptomic analysis showing the increase of hypertrophic growth related markers are shown.
Below are my points, listed as major and minor, which may help the Authors reworking their manuscript making.
Major
- In the Introduction the Authors report that: “we aimed to determine whether YB-1 is modified in cardiac patients who develop heart failure”, so why the Authors have chosen rat cells as cellular model instead of human cells for further experiments?
I understand that data produced in rats and mice have been published (as the Authors state in the Introduction), but given what the Authors report in Figure 1, which gives a strong starting point, the Authors should have performed experiments in human cells, using rat cells as positive control.
Regarding the Figure 1, considering the impressive reduction shown in WB in Figure 1A, I suggest the Authors to show YB-1 immunoblots of all 8 patients they analyzed. It would be really interesting to the readers to “see” how consistent and conserved the reduction between patients is. The Authors might consider keeping the figure as it is in the current version while adding the WB for all patients in a supplementary figure.
I am not convinced by protein densitometry shown in Figure 1B; the difference in YB-1 protein amount between control and ICM samples is much greater than that shown in the graph. There is no information in the Materials and Methods section about densitometry; how was it done?
Furthermore, molecular weights are missing and original WB images for this experiment are not provided.
- The significance of Figure 2 is really poor.
The Authors basically show technical data about the efficiency of KD (which can be moved in a supplementary figure) and do not stress properly the point anticipated in the title.
The y-axis of panel E is wrong, cellular area cannot be 4000-8000 mm2; I assume that they plotted the number of the cells analyzed (as the numbers in the graph resemble the “n” reported in the caption) rather than cellular dimension. How was this analysis made? What are the Authors actually plotting in panel E?
The Authors report in the Materials and Methods sections that: “The size and area of H9C2 cells were determined 72 hours post-transfection with YB-1 or control siRNA cultured in media with 1% FCS.”, but this is not sufficient. Furthermore, scale bar is missing in panels F, G and H and the image in panel H is really of poor quality, making difficult to draw conclusions. Maybe a higher magnification of control and KD cells would help the Authors to make this point clear, but there is no indication that the increase of cellular area in YB-1 KD is hypertrophic growth and not siRNA induced size alteration (well documented in literature) which has nothing to do with hypertrophic growth. Why the Authors did not perform a control experiment (PE or GDF15 treatment as they did in the experiments shown in Figure 7)? With these premises the title of the paragraph is misleading.
Moreover, in the panel C the Authors must name the lanes of the western blot. The original images of this WB are missing in the file “original images”.
The y-axis upper limit of the RT-qPCR panels should be the same. No indication of the “n” is reported for 72h KD; no indications for what “n” stands for (see the point about statistics).
The Authors reported in the Materials and Methods section that data are analyzed as
2-DDCt, so why are they plotting data as “percentage of control” with SEM? Why are they showing a further normalization which is not described in the text? They should plot either DCt or 2-DDCt or also fold-change values with Confidence intervals (CI). Furthermore, the Authors performed Reverse Transcription quantitative real Time PCR (RT-qPCR) and not Real Time PCR (RT-PCR), so they must address it properly in the text.
- The increase in protein synthesis shown in Figure 3, does not mean per se hypertrophic growth. The Authors must add data (for instance immunohistochemical analysis, western blot or transcriptomic analysis) in which the increased expression of hypertrophic growth markers is visible.
- The effect of the LY294002 is really low. Perhaps the scale on the y-axis does not help.
However why have the Authors chosen LY294002 which is widely known as non-specific PI3K inhibitor instead of other commercially available PI3K specific inhibitors?
- In the caption of Figure 6 the Authors report that panel A represents a RT-qPCR while panel B (lower part) represents a western blot quantification; how is it possible that they look exactly the same in terms of scale and x/y axis? Why is the RT-qPCR notation so variable in the whole manuscript? In this figure the y-axis notation is “x-fold to normalized controls”, while previously it was “percentage of controls”, please be consistent.
The images shown in Figure 6C-F make really difficult for the reader to observe what the Authors describe as “YB-1’s distinct structure”. With this magnification and resolution is impossible to observe sarcomeres.
- The effect of YB-1 overexpression thorough infection prior PE treatment is not convincing. Have the Authors performed statistical analysis on PE-AdYB-1 vs PE-Control?
In my opinion the Statistical Analysis section is not clear and is incomplete.
It seems that the Authors used the same set of analysis and choose the same way of data presentation for all the experiments, but this is incorrect.
There is no indication about normality tests used. In the figure captions “n” is reported, but it is not indicated anywhere in the text if it refers to the number of biological or technical replicates.
The sentence: “Wherever error bars are seen, the data have been normalized to an averaged control value” sounds strange, the Authors should replace it with “data are presented as percentage of control”.
When the Authors state that “Statistical comparisons used one-way analyses of variance or the Student–Newman–Keuls test for post hoc analyses” it sounds that it applies to all the analysis they made, but this is not the case (at least I assume that). For example, in Figure 1B a simple an unpaired two-tailed t-test (with or without Welch’s correction) or a Mann-Whitney (depending on normality of the samples) should be used.
In RT-qPCR confidence intervals rather than SEM should be used It is not clear on which sample (and how) statistical analysis have been performed. The absence of statistical notation on some samples (Figure 7 “PE-AdYB-1 vs PE-Control”) makes difficult to understand if the analyses have not been performed or the results were not significant.
Minor
- In section 2.2 lines from 114 to 120 could be summarized in one sentence in which the Authors report that YB-1 KD was proficient in protein reduction already after 24 hours. The same could be said for section 2.3 lines 137-140.
- Why using an anti-human YB-1 for western blot on mouse/rat extract while there are commercially available antibodies against those species?
- The anti-YB-1 product code is “ab12148” and not “ab122148”, please rectify.
Comments on the Quality of English Language
Minor editing of English language required.
Author Response
Reviewer 2
We thank the editor and the reviewers for their time and their constructive critiques and suggestions to improve our manuscript. We have addressed each of the comments and have revised the manuscript accordingly. We hope that the editor and reviewers will now find the revised manuscript acceptable for publication.
Major
In the Introduction the Authors report that: “we aimed to determine whether YB-1 is modified in cardiac patients who develop heart failure”, so why the Authors have chosen rat cells as cellular model instead of human cells for further experiments?
I understand that data produced in rats and mice have been published (as the Authors state in the Introduction), but given what the Authors report in Figure 1, which gives a strong starting point, the Authors should have performed experiments in human cells, using rat cells as positive control.
The proposal to carry out the tests in human cell lines has advantages and disadvantages. Human cells like AC16 are proliferating human cardiomyocytes. Cardiomyocytes of adult myocardium increase their cellular mass in response to growth stimuli. They undergo hypertrophic growth but they do not proliferate in contrast to immature cardiomyocytes. Thus, human cell lines are not significantly better than H9C2 cells as controls.
Therefore, we prefer a cellular model that shows hypertrophic growth like adult rat cardiomyocytes instead of proliferating human cell lines. Although it is a different species, we believe that this cellular model resembles the in vivo situation in humans much closer. Our finding that YB-1 is downregulated in ICM patients indicates that this may play a pathophysiological role in humans.
Regarding the Figure 1, considering the impressive reduction shown in WB in Figure 1A, I suggest the Authors to show YB-1 immunoblots of all 8 patients they analyzed. It would be really interesting to the readers to “see” how consistent and conserved the reduction between patients is. The Authors might consider keeping the figure as it is in the current version while adding the WB for all patients in a supplementary figure.
As suggested, we add the western blot for all patients in a supplementary figure, and modified figure 1 (see next question).
I am not convinced by protein densitometry shown in Figure 1B; the difference in YB-1 protein amount between control and ICM samples is much greater than that shown in the graph. There is no information in the Materials and Methods section about densitometry; how was it done?
Furthermore, molecular weights are missing and original WB images for this experiment are not provided.
Quantification of western blots were performed with Quantity One software (Biorad) using volume tools for volume analysis. The background was subtracted and the volume of the bands of YB-1 was set to the volume of the bands of the reference protein. For the presentation, all individual values were related to the mean value of the controls. The presentation is shown as a percentage. This is now also mentioned in the methods part line 580-584.
We have modified Figure 1 according to the suggestions of reviewer 1. The labeling was changed from A and B to A (i) and A (ii). We have chosen a different location in the original blot to display the representative blot, which better reflects the different expression levels between CON and ICM. We added the molecular size for YB-1 and GAPDH in figure A (i). Furthermore, the complete western blot of all human samples is now presented in supplement figure S1.
The significance of Figure 2 is really poor.
The Authors basically show technical data about the efficiency of KD (which can be moved in a supplementary figure) and do not stress properly the point anticipated in the title. The y-axis of panel E is wrong, cellular area cannot be 4000-8000 mm2; I assume that they plotted the number of the cells analyzed (as the numbers in the graph resemble the “n” reported in the caption) rather than cellular dimension. How was this analysis made? What are the Authors actually plotting in panel E?
Digital images of the cells were obtained using a microscope and BZ Observation Software® at a 100- 200x magnification. Cell area was analyzed using the BZ Analyser Software® by framing each cell. The analysis software automatically converted the cell area. For the measurement of hypertrophic growth under YB-1 siRNA, the increase in cell area compared to controls is particularly important. Therefore, we have decided not to give any absolute value; we now show the changes in cell area as a percentage.
The Authors report in the Materials and Methods sections that: “The size and area of H9C2 cells were determined 72 hours post-transfection with YB-1 or control siRNA cultured in media with 1% FCS.”, but this is not sufficient.
H9C2 cells were grown in Dulbecco’s modified Eagle medium supplemented with 10% FCS, 1% penicillin/streptomycin, 100 mM sodium pyruvate, and 200 mM glutamate at 37°C with 5% carbon dioxide. For the planned experiments, H9C2 cells were split and plated onto several small dishes. H9C2 cells were cultured for 24 hours before transfection with YB-1 siRNA (SI01921591; Qiagen, Hilden, Germany;.target sequence: 5’-ACC AAG GAA GAC GTA TTT GTA-3’; sense: 5’-CAA GGA AGA CGU AUU UGU ATT-3’; antisense: 5’-UAC AAA UAC GUC UUC CUU GGT-3’) or AllStars negative control siRNA (1027281, Qiagen, Hilden, Germany; sequences not disclosed by the manufacturer) using Lipofectamine 2000 (Invitrogen, Carlsbad, CA, USA). After 24 hours, cells were transfected with siRNA according to the manufacturer’s instructions in medium without FCS. After eight hours of incubation, the medium was replaced with Dulbecco’s modified Eagle medium supplemented with, 1% penicillin/streptomycin, 100 mM sodium pyruvate, and 200 mM glutamate and 1 % FCS. Cells were cultured at 37°C with 5% carbon dioxide for 72 hours. Then cell size was determined. Digital images of the cells were obtained using a microscope and BZ Observation Software® at a 100- 200-x magnification. Cell area was analyzed using the BZ Analyser Software® by framing each cell. The analysis software automatically converted the cell area. Changes in cell area were calculated as a percentage compared to controls. This detailed description is now mentioned in the method part in line 471-488.
Furthermore, scale bar is missing in panels F, G and H and the image in panel H is really of poor quality, making difficult to draw conclusions. Maybe a higher magnification of control and KD cells would help the Authors to make this point clear, but there is no indication that the increase of cellular area in YB-1 KD is hypertrophic growth and not siRNA induced size alteration (well documented in literature) which has nothing to do with hypertrophic growth.
We have chosen a better quality image. Figure 2H is now Figure 2G. As explained in the next answer (see below), we believe that we don’t see siRNA-induced cell size alterations by YB-1 siRNA.
Why the Authors did not perform a control experiment (PE or GDF15 treatment as they did in the experiments shown in Figure 7)? With these premises the title of the paragraph is misleading.
We did perform experiments with PE. This figure is now added in the supplement and the data are mentioned in the result section.
As you can see in the figure, hypertrophy induction of PE is very low. The hypertrophy stimulus with 10 µM PE is very low for H9C2 cells. In the literature, concentrations of 50 or 100 µM PE are more common. However, we used the same concentration of PE as in adult cardiomyocytes. Furthermore, the Ctrl siRNA has a similar increase in the cell size as PE. This would allow us to assume that the increase in cell size of the Ctrl siRNA is a siRNA-induced size alteration. In contrast to Ctrl siRNA, YB1 siRNA shows a much greater increase in cell size than PE. This increase is significant towards PE and Ctrl siRNA, whereas there is no change in cell size between YB-1 siRNA and YB-1 siRNA and PE. This indicates that the increase in cell sizes under YB-1 siRNA is due to hypertrophic growth.
Moreover, in the panel C the Authors must name the lanes of the western blot. The original images of this WB are missing in the file “original images”.
Figure 2 C is now figure 2 B (i). The bands are arranged so that they are positioned exactly above the respective bar of the quantitative evaluation. Furthermore, the labeling has been supplemented.
We added all original blots of figure 2 C in the file “original images”.
The y-axis upper limit of the RT-qPCR panels should be the same. No indication of the “n” is reported for 72h KD; no indications for what “n” stands for (see the point about statistics).
As you requested, we have displayed the real-time qRT-PCR results as mean ± CI. To have the same y-axis, we combined figure 2A and 2B with the implemented statistical analysis between the different points. In figure 2 A we now show the combined data from 24 and 48 hours of siRNA transfection.
Figure 2. YB-1 downregulation in H9C2 cells resulted in hypertrophic growth. H9C2 cells were transfected with YB-1 or Ctrl siRNA or were non-transfected. (A) Their mRNA levels were quantified after 24 hours (n = 4) or 48 hours (n = 8) with real-time qRT-PCR using HPRT1 as the housekeeping gene. Data are presented as the means ± CI. Key: *, p < 0.05 versus control after 24 hours; #, p < 0.05 comparison as indicated.
Unless otherwise stated, all n-numbers are biological replicates. This is now implemented in the statistic section line 604.
As mentioned in the figure legend of figure 2, YB-1’s protein levels were examined using western blots (n = 4) after 72 hours using vinculin as the internal control.
The Authors reported in the Materials and Methods section that data are analyzed as
2-DDCt, so why are they plotting data as “percentage of control” with SEM? Why are they showing a further normalization which is not described in the text? They should plot either DCt or 2-DDCt or also fold-change values with Confidence intervals (CI). Furthermore, the Authors performed Reverse Transcription quantitative real Time PCR (RT-qPCR) and not Real Time PCR (RT-PCR), so they must address it properly in the text.
We performed a Real-Time Quantitative Reverse Transcription PCR, which we have abbreviated as real-time RT-PCR, and RT stands for reverse transcription. We now added the q in the text and used the following spelling “real-time qRT-PCR” (www.ncbi.nlm.nih.gov/probe/docs/techqpcr/).
Originally, data were first calculated as dCt and than as 2-ddCt. To better illustrate this, the controls were converted to 100 % and this normalization was shown. Now, we followed the reviewer's suggestion: In the revised version, we plot the real-time qRT-PCR results as 2-ddCt expressed as mean ± confidence intervals (CI).
The increase in protein synthesis shown in Figure 3, does not mean per se hypertrophic growth. The Authors must add data (for instance immunohistochemical analysis, western blot or transcriptomic analysis) in which the increased expression of hypertrophic growth markers is visible.
The protein synthesis measurement was carried out by means of incorporation of 14C-phenylalanine since the amino acid phenylalanine is almost exclusively used for protein synthesis and is not metabolized by cardiomyocytes. The rate of protein synthesis increase in a cell during hypertrophic growth can be higher compared to normal cells, so an increased rate of protein synthesis qualitatively indicates hypertrophy. As another parameter for hypertrophic growth, we now additionally determined cell area of adult cardiomyocytes. These measurements are now displayed in figure 3 (figure 3E, see below) that resembles the results of the protein synthesis rate. Together with an enlargement of the cells, this is a clear indication of hypertrophic growth.
Since ventricular cardiomyocytes were isolated, not all markers for hypertrophy are expressed (BNP ventricle, ANP atria). In addition, the timing of the analysis determines when which protein is expressed. BNP is an early response gene. At the times when we isolated RNA, we could no longer detect an increase in BNP, since BNP mRNA is rapidly degraded through a translation-dependent destabilization mechanism in cultured ventricular cardiomyocytes (Nakagawa et al., JCI 1995, 1280-1287).
- The effect of the LY294002 is really low. Perhaps the scale on the y-axis does not help.
However why have the Authors chosen LY294002 which is widely known as non-specific PI3K inhibitor instead of other commercially available PI3K specific inhibitors?
We have changed the y-axis. As we had already worked with Ly294002 in earlier studies and tested the concentrations for various questions, we decided to continue with this inhibitor.
In the caption of Figure 6 the Authors report that panel A represents a RT-qPCR while panel B (lower part) represents a western blot quantification; how is it possible that they look exactly the same in terms of scale and x/y axis? Why is the RT-qPCR notation so variable in the whole manuscript? In this figure the y-axis notation is “x-fold to normalized controls”, while previously it was “percentage of controls”, please be consistent.
We are not sure what the reviewer is actually pointing at. The axis do not look the same: figure A shows six samples and the time points are in between 24 hours, whereas B shows seven samples and the time points are in between 36 hours. The YB-1 expression gets stronger as time goes on. This applies to both the RNA and the protein. The control virus is shown at the end for comparison in both cases.
Since the overexpression is very strong, we used x-fold instead of percent. We have now changed this and use percentage as for all other western blot figures.
Furthermore, according to the suggestion of the reviewer, that presentation and calculation of data should be the same, we now present 2-ddCt for real-time qRT-PCR and western blot data in % change, as has been done in all other figures.
The images shown in Figure 6C-F make really difficult for the reader to observe what the Authors describe as “YB-1’s distinct structure”. With this magnification and resolution is impossible to observe sarcomeres.
We have changed figure 6 and added enlarged section of the cardiomyocytes with better resolution, so that the sarcomeres are now clearly visible.
The effect of YB-1 overexpression thorough infection prior PE treatment is not convincing. Have the Authors performed statistical analysis on PE-AdYB-1 vs PE-Control?
Many thanks for this hint. We have carried out a statistical analysis in relation to PE and added rhombuses as signs for significance.
In my opinion the Statistical Analysis section is not clear and is incomplete.
It seems that the Authors used the same set of analysis and choose the same way of data presentation for all the experiments, but this is incorrect.
There is no indication about normality tests used. In the figure captions “n” is reported, but it is not indicated anywhere in the text if it refers to the number of biological or technical replicates.
The sentence: “Wherever error bars are seen, the data have been normalized to an averaged control value” sounds strange, the Authors should replace it with “data are presented as percentage of control”.When the Authors state that “Statistical comparisons used one-way analyses of variance or the Student–Newman–Keuls test for post hoc analyses” it sounds that it applies to all the analysis they made, but this is not the case (at least I assume that). For example, in Figure 1B a simple an unpaired two-tailed t-test (with or without Welch’s correction) or a Mann-Whitney (depending on normality of the samples) should be used.
The information about the statistics has been supplemented and now reads as follows in the method section page 18: “All n-numbers are biological replicates. The real-time qRT-PCR was performed in duplicates and the mean value was used for the calculation. The results are expressed as the mean ± standard error of the mean (SEM) or as mean ± confidence intervals (CI) for real-time qRT-PCR. All variables were evaluated for normal distribution using the Kolmogorov–Smirnov and Shapiro-Wilk tests. Levene’s test was used to control for variance of homogeneity. If two variables were compared directly, an unpaired two-tailed t-test or a Mann-Whitney (depending on normality of the samples) was used. For more than two variables a one-way analysis of variance (ANOVA) was used; followed by Student–Newman–Keuls test for post hoc analyses if suitable. All results with p < 0.05 were considered statistically significant. All data analyses were performed with the SPSS software (version 27; SAS Institute Inc., Cary, NC, USA).”
In RT-qPCR confidence intervals rather than SEM should be used It is not clear on which sample (and how) statistical analysis have been performed. The absence of statistical notation on some samples (Figure 7 “PE-AdYB-1 vs PE-Control”) makes difficult to understand if the analyses have not been performed or the results were not significant.
For real-time qRT-PCR the results are now expressed as means ± confidence intervals (CI).
Many thanks for pointing out the absence of statistical notations in some samples. We have added the signs for significance where they were still missing. To come back to the example you mentioned, the two samples PE control and PE AdYB-1 differ significantly and underline the inhibition of PE induction by YB-1.
Minor
- In section 2.2 lines from 114 to 120 could be summarized in one sentence in which the Authors report that YB-1 KD was proficient in protein reduction already after 24 hours. The same could be said for section 2.3 lines 137-140.
Neither in section 2.2 nor in section 2.3 can it be summarized that the protein expression has already decreased significantly after 24 hours. In section 2.2 we show reduction in RNA expression after 24 and 48 hours and reduction in protein expression after 72 hours. In section 2.3 we show reduction in RNA expression after 48 hours and reduction in protein expression after 48 and 72 hours. As can be seen, RNA expression is significantly lower than protein expression at the given time. For these reasons, we could not summarize these points.
- Why using an anti-human YB-1 for western blot on mouse/rat extract while there are commercially available antibodies against those species?
When we started with this project some years ago, we have chosen this antibody, which was predicted to work with mouse and rat. We tested it and it worked. Then we stayed with it. Finally, there are also other publications that used this antibody for non-human samples as well (for example: PMID: 26725322, PMID:28784777).
- The anti-YB-1 product code is “ab12148” and not “ab122148”, please rectify.
Thank you for the hint. We corrected the spelling mistake.
The PDF file also contains figures.

Round 2
Reviewer 2 Report
Comments and Suggestions for Authors
I wish to thank the Authors for the effort put in revising the manuscript.
They clarified all my doubts and addressed properly all my concerns.
With the integrations and modifications they made the manuscript has improved a lot and I definitely recommend the present version for publication in IJMS.
Minor editing of English language required.